# Deep-Learning-Based Automatic Detection and Segmentation of Brain Metastases with Small Volume for Stereotactic Ablative Radiotherapy

**DOI:** 10.3390/cancers14102555

**Published:** 2022-05-23

**Authors:** Sang Kyun Yoo, Tae Hyung Kim, Jaehee Chun, Byong Su Choi, Hojin Kim, Sejung Yang, Hong In Yoon, Jin Sung Kim

**Affiliations:** 1Department of Radiation Oncology, Yonsei Cancer Center, Yonsei University College of Medicine, Seoul 03722, Korea; skyunyoo@yuhs.ac (S.K.Y.); thkim@eulji.ac.kr (T.H.K.); cjhsmile@yuhs.ac (J.C.); cbs1009@yuhs.ac (B.S.C.); hjhenrykim@yuhs.ac (H.K.); 2Medical Physics and Biomedical Engineering Lab (MPBEL), Yonsei University College of Medicine, Seoul 03722, Korea; 3Department of Radiation Oncology, Nowon Eulji Medical Center, Eulji University School of Medicine, Seoul 01830, Korea; 4Oncosoft Inc., Seoul 03787, Korea; 5Department of Biomedical Engineering, Yonsei University, Wonju 26493, Korea; syang@yonsei.ac.kr

**Keywords:** brain metastases, magnetic resonance imaging, autosegmentation, deep learning, convolutional neural network, stereotactic ablative radiotherapy

## Abstract

**Simple Summary:**

With advances in radiotherapy (RT) technique and more frequent use of stereotactic ablative radiotherapy (SABR), precise segmentation of all brain metastases (BM) including a small volume of BM is essential to choose an appropriate treatment modality. However, the process of detecting and manually delineating BM with small volumes often results in missing delineation and requires a great amount of labor. To address this issue, we present a useful deep learning (DL) model for the detection and segmentation of BMwith contrast-enhanced magnetic resonance images. Specifically, we applied effective training techniques to detect and segment a BM of less than 0.04 cc, which is relatively small compared to previous studies. The results of our DL model demonstrated that the proposed methods provide considerable benefit for BM, even small-volume BM, detection, and segmentation for SABR.

**Abstract:**

Recently, several efforts have been made to develop the deep learning (DL) algorithms for automatic detection and segmentation of brain metastases (BM). In this study, we developed an advanced DL model to BM detection and segmentation, especially for small-volume BM. From the institutional cancer registry, contrast-enhanced magnetic resonance images of 65 patients and 603 BM were collected to train and evaluate our DL model. Of the 65 patients, 12 patients with 58 BM were assigned to test-set for performance evaluation. Ground-truth for BM was assigned to one radiation oncologist to manually delineate BM and another one to cross-check. Unlike other previous studies, our study dealt with relatively small BM, so the area occupied by the BM in the high-resolution images were small. Our study applied training techniques such as the overlapping patch technique and 2.5-dimensional (2.5D) training to the well-known U-Net architecture to learn better in smaller BM. As a DL architecture, 2D U-Net was utilized by 2.5D training. For better efficacy and accuracy of a two-dimensional U-Net, we applied effective preprocessing include 2.5D overlapping patch technique. The sensitivity and average false positive rate were measured as detection performance, and their values were 97% and 1.25 per patient, respectively. The dice coefficient with dilation and 95% Hausdorff distance were measured as segmentation performance, and their values were 75% and 2.057 mm, respectively. Our DL model can detect and segment BM with small volume with good performance. Our model provides considerable benefit for clinicians with automatic detection and segmentation of BM for stereotactic ablative radiotherapy.

## 1. Introduction

Brain metastases (BM) are 10 times more common than primary brain tumors and occur in about 20% of all patients with cancer [1]. Recent advances in diagnostic methods and systemic therapies have improved the survival of cancer patients [2]. The increased detection of BM has made their treatment extremely important, more so than even previous cancer treatment [3,4].

Treatment options for BM, including surgery and radiotherapy (RT), vary depending on their size, number, and location. Traditionally, whole-brain RT is considered the treatment of choice for numerous BM [5], but recent advances in RT techniques introduced stereotactic ablative radiotherapy (SABR) for BM. Benefiting from efficient local control [6], SABR is becoming one of the major advances in the treatment of BM [7]. Therefore, early detection and accurate segmentation of BM are crucial for appropriate treatment planning. Contrast-enhanced magnetic resonance imaging (CE-MRI) is known as the imaging modality of choice for patients with suspected BM. However, even with MRI, detecting and manually delineating BM are labor-intense processes [8]. Even for experienced experts, BM are occasionally mistaken for a normal brain, resulting in missing delineation, especially when its volume is small.

Recently, automatic segmentation of brain tumors or BM using deep learning (DL) has been regarded as a promising approach [9]. In 2015, Losch, M et al. [10] published the first study for the segmentation of BM with DL algorithms. The results of the first study were 0.66 Dice coefficient, 82.8% sensitivity, and the average number of false positive was 7.7 per patient. Since then, there have been several efforts [11,12,13,14,15] to develop the DL algorithms assisting clinical practice by autodetect BM and perform segmentation. The cascade three-dimensional (3D) fully convolutional networks for BM detection and segmentation (BMDS net) [11] was used for BM detection and segmentation. The BMDS net detected all BM in 1652 patients with a detection accuracy of 100%; however, the mean volume of BM was 4.01 cc, which was relatively larger than the mean volume in our study. While most studies using DL algorithms incorporated MRI with T1Gd, O. Charron et al. [13] used different MRI modalities simultaneously. In their study, the convolutional neural networks (CNNs) detected and segmented BM better when using both T1Gd and T2 or fluid-attenuated inversion recovery (FLAIR) images than with T1Gd alone. The mean volume of BM in their study was 2.4 cc.

However, these studies have dealt with BM with relatively large volumes, and few studies have described [12], detected, and segmented BM with small volume that pose a challenge to physicians in clinical practice. The cases of large-volume BM and small-volume BM are showed in Figure 1.

In this study, we developed an advanced DL-based model that is more useful in clinical applications to BM detection and segmentation in CE-MRI by testing a relatively smaller volume than in previous studies.

## 2. Materials and Methods

### 2.1. Dataset

A list of patients with BM who were treated with RT between 2016 and 2020 was extracted from the institutional cancer registry. The clinical data and MRI findings of 65 patients were collected. All MRI consisted of T1 gadolinium-enhanced images (T1Gd). One expert radiation oncologist (TH.K.) delineated the ground-truth BM on MRI using MIM software (Mim Software Inc., Cleveland, OH, USA), and another radiation oncologist (HI.Y.) cross-checked the ground-truth BM.

The median age of the patients was 63 years (range, 19–87), with a predominance of male patients (54%, *n* = 35; *n* is number). The most common primary cancer was lung (86%, *n* = 56).

Since each magnetic resonance (MR) image had a different resolution, pixel spacing was resampled to 0.195, the best resolution in our dataset. Except for one data with a slice thickness of 3 mm, all images had a slice thickness of 1 mm. Images with 3-mm slice thickness were resampled to 1 mm. As a result, all patient images had a resolution of 0.195 × 0.195 × 1 mm${}^{3}$, with a size of 1024 × 1024.

The dataset used in our study consists of the T1Gd, which is the input of the model, and manual delineation, which is the label of the model. In addition, the dataset was divided into the train(+valid) set, which is for finding the optimal parameters of the model, and the test set. The train(+valid) set was used to train the model, and the test set was used to evaluate the performance of the model.

### 2.2. Brain Metastases Populations

Data for 603 BM were collected, of which 58.2% had a volume of <0.1 cc. Among them, 41.2% had a volume of <0.04 cc. Based on the 1 mm slice thickness resampled image, each BM occupied an average of 5 slices. Each patient had a different number of BM (range 1–117; the volume characteristics of patient with up to 117 BM are: The whole-brain volume is about 1.2 × 10${}^{6}$ cc, maximum and minimum volumes are 0.226 cc and 0.004 cc, respectively, and mean volume is 0.037 cc.).

Of the 65 patients, data for 12 patients were randomly selected and used as unseen data (test-set) to evaluate the model’s performance. In the test-set, 41.2% BM had a volume of <0.04 cc. The following are the characteristics of BM in the test-set: The maximum and minimum volumes are 1.219 cc and 0.021 cc, respectively, and median and mean volumes are 0.068 cc and 0.158 cc, respectively. Details of patient characteristics and BM are summarized in Table 1.

### 2.3. Deep Learning Strategy

Unlike Hounsfield Units for computed tomography images, MR images do not have a consistent intensity index, and this leads to inconsistencies in the signal intensity and volume between images of different patients [16]. As a preprocessing to deal with these inconsistencies, N4 bias field correction [17] was applied to compensate for the corrupted bias, and gamma correction [18] was applied.

The size and location of BM were different; these differences created the problems of unbalanced data and small volumes in DL training. Therefore, we cropped the original slice size (1024 × 1024 size) into overlapping patches (128 × 128 size) and applied the overlapping patch technique, an effective method to extract small-volume BM from the normal brain that occupies most of the slice [19,20]. The overlapping patch technique divides regions by sliding a patch of a certain size in on slice by a certain interval. In our study, a 128 × 128 size patch on a 1024 × 1024 size slice was sliding from left to right of the slice and from top to bottom at an interval of 64 pixels to form a cropped patch. We have tested a few different value (512 × 512, 256 × 256, 128 × 128, 64 × 64 size) for patch size but have not done an exhaustive investigation. However, we found that the results were optimal in 128 × 128 size. As the size of the cropped patch is small, it is advantageous to learn about the smaller BM, but when the size of the patch is too small (64 × 64 size in our case), it is difficult to learn the structure of the region not including the BM. For the interval of 64 pixels to form the cropping patch, in order for all adjacent patches to include the patch’s partial information, divided into a grid of half the size (64 pixels) of patch.Because the number of data in the healthy tissue samples and tumor samples was imbalanced, we applied the undersampling technique [21,22] in each batch randomly. Therefore, the number of healthy data was reduced, and the number of BM samples was increased, so that the data numbers of healthy and tumor samples are balanced during training.

To avoid the inefficiencies of a 3D architecture and to utilize volume information, we conducted 2.5-dimensional (2.5D) training using the two-dimensional (2D) U-Net [23] as the DL architecture for the detection and segmentation of BM. CNNs typically process color images consisting of three input channels (RGB; red, green, blue). In our 2.5D training, this setup was designed by allocating five consecutive axial slices to extended five input channels. In this 2.5D training, we performed the 2.5D overlapping patch technique by composing the overlapping patches instead of the entire axial slices.

After training, in each cropped patch, and outputs the probability value. When the cropped patch is merged back to the original image size (1024 × 1024 size), these probability values are averaged to detect and segment the entire tumor.

Our deep learning strategy for effectively detecting and segmenting BM with small volumes is depicted in Figure 2.

### 2.4. Deep Learning Details

Prior to training, all images were applied with intensity scaling within the range of 0.5–0.95% after an N4 bias field correction.

A scaled image of 1024 × 1024 size was cropped to 128 × 128 size using the overlapping patch technique. The overlapping patch technique divides regions by sliding a patch of a certain size in on slice by a certain interval. In our study, a 128 × 128 size patch on a 1024 × 1024 size slice was sliding from left to right of the slice and from top to bottom at an interval of 64 pixels to form a cropped patch. Even if a 1024 × 1024 size image with 0.195 pixel spacing is cropped into a 128 × 128 size image, pixel spacing does not change. The overlapping patch technique does not assume that the entire tumor will be contained within each patch. If the patch does not contain any tumor, the DL learns from the patch about the area that should not be segmented. Conversely, if the patch contains the entire tumor, the DL can learn a smaller BM from the patch, and if the patch contains a portion of the tumor, the DL can learn a relatively large BM. Our DL model detects and segments the corresponding area if all or part of the tumor is included in each cropped patch, and outputs the probability value. When the cropped patch is merged back to the original image size, these probability values are averaged to detect and segment the entire tumor. Therefore, while having advantages for small-volume tumors, it is possible to give satisfactory detection and segmentation results even for tumors larger than the patch size.

To increase the diversity of data, after applied undersampling technique, data augmentations (horizontal/vertical flip, random rotation/blur) applied randomly on the fly in each batch. When the 2.5D overlapping patch technique was applied, the number of healthy patches was 428,159 while the number of tumor patches (in case of partially included or entirely included) was 23,866. After the undersampling technique was applied, the final healthy patches and tumor patches was 30,000 each.

The U-Net was implemented using Python 3.8.3 (Python Software Foundation, Beaverton, OR, USA) and PyTorch 1.5.1 [24]. The U-Net consists of a contracting encoder and an expanding decoder. The contracting encoder extracts low-level features, and the expanding decoder, which enables precise localization, produces the label map. In contracting encoder of standard U-Net, each layer contains two 3 × 3 convolutions, each followed by rectified linear unit (ReLU) and a 2 × 2 max pooling operation with stride 2 for downsampling. In expanding decoder of standard U-Net, consists of 2 × 2 convolutions for upsampling and two 3 × 3 convolutions, each followed by a ReLU. To avoid the problem of gradient becoming 0, exponential linear unit (ELU) [25] was applied instead of ReLU and Batch Normalization was applied before ELU.

The U-Net had an initial learning rate of 0.0001 and was trained by reducing the learning rate by 0.5 times when a metric stopped improving every 2 epochs until 30 epochs. We set the batch size to 256, which was the limit of the GPU memory. The DL architecture’s weights were optimized using Adam [26] and L2-regularization (β = 0.00001).

### 2.5. Statistical Analysis

The performance of our study was evaluated both on detection and segmentation. The sensitivity and average false positive rate were taken as measurements for detection performance. The definition of sensitivity metrics is given below: (1)Sensitivity=TP/(TP+FN)
where true positive (TP) and false negative (FN) denote the number of BM correctly detected and the number of undetected BM, respectively. The false positive (FP) means that a non-BM was incorrectly detected as BM. Results of FP were given as an average of the results on the 12 patients of the test-set. The definition of average false positive rate is given below: (2)$$Average\;False\;Positive\;Rate=\sum_{i=1}^{n}FP_{i}/N$$
where FP${}_{i}$ is the number of incorrectly detected as BM in each patient in the test-set and *N* denote the number of patients in test-set.

The Dice coefficient [27] (DICE), Dice coefficient with dilation (DWD) and the 95% Hausdorff distance (HD95) [28] were taken as measurements for segmentation performance. The DICE, which is generally used as a metric of segmentation performance, is difficult to apply to small-volume BM because it shows a large difference even if the predicted result deviates from the ground truth only slightly. Recent studies question of the correlation of commonly used measures such as the DICE and clinical acceptability. There was an opinion that when evaluating performance, it should be evaluated with a different meaning depending on the goal [29]. When SABR is performed clinically, the delineation is modified by giving a margin of 2–3 mm. Therefore, in this study, we mainly used DWD to evaluate the segmentation performance. The DWD, the value obtained by dilation of the prediction (PR) and the dilation of the ground truth (GT). A margin of 3 mm does not affect local control [30], so we dilate both prediction and ground truth to 3 mm. The definitions of these metrics are given below: (3)DICE=(2*|GT∩PR|)/(|GT|+|PR|)
(4)$$DICE\;with\;dilation=(2*|GT^{\prime }\cap PR^{\prime }\left|\right)/(|GT^{\prime }|+|PR^{\prime }\left|\right)$$
where GT${}^{\prime }$ is the dilation of the ground truth with 3 mm and PR${}^{\prime }$ is the dilation of the prediction with 3 mm. The HD95 was defined as the 95 percentile distance over all point distance in PR to its closest point in GT. The definition of this metric is given below: (5)$$HD95=95\%(min_{gt\in GT}d\left(pr,gt\right))\forall pr\in P$$
where d(pr,gt) is the distance between the points pr and gt in PR and GT, respectively.

## 3. Results

### 3.1. Detection Performance

A performance evaluation was conducted on 58 BM in the MRI of 12 patients. Of the total of 65 patients, 53 were assigned to the Train(+Valid) set and 12 were assigned to the Test-set. Our proposed model detected all BM over 0.04 cc and showed a sensitivity of 85.7% for BM less than 0.04 cc. For the overall test-set, 56 out of 58 BM were detected, showing a sensitivity of 96.9%. In each patient, up to 6 and at least 0 false positives were found, and the average number of false positives was 1.25 per patients. Some examples of detection and false positives are shown in Figure 3.

The locations of false positives can be summarized in three ways (Figure 4). First, the false positives were found outside the brain structure. Second, the false positives were found in a structure with high intensity. Third, the false positives were found in the superior sagittal sinus.

Table 2 summarizes the DL performance according to the number of metastases in each patient in the test-set before rejecting any structure with a volume of <0.02 cc. Out of the test-set, for 9 patients (75%) with a BM of less than 10, detection sensitivity was 100%. For 3 patients (25%) with 10 or more BM, the detection sensitivity was 93.5%; it was 90% and 100% for patients with 10 and 11 metastases, respectively.

### 3.2. Segmentation Performance

The DL performance for test-set BM segmentation is summarized in Table 3. The Dice coefficient showed a value of 55% for all BM. Our test-set consisted of small-volume BM, mostly <0.1 cc, and even a slight deviation from the prediction showed a large difference in the DICE. To compensate for this, DWD and HD95 were used as the segmentation performance metric. In the BM with a volume <0.04 cc, DWD and HD95 showed values of 63.2% and 1.689 mm except for those that were undetected, respectively, and the BM with a volume >0.04 cc showed values of 78.6% and 2.158 mm, respectively. For all BM, the DWD value was 75% and the HD95 value was 2.057 mm. Some examples of segmentation are shown in Figure 5.

## 4. Discussion

In this study, we demonstrated that gross tumors can be detected and segmented by the developed advanced DL modelthat usage of graphical processing units (GPUs) (NVIDIA TITAN RTX with 24 GB of memory, NVIDIA CORPORATE, Santa Clara, CA, USA). In total, 43,200 patches by applying the 2.5D overlapping patch technique from 12 patients in test-set were used to inference the trained model.

The final model used in our study had an overall sensitivity of 97%. We found that the performance of algorithms was reduced for lesions <0.06 cc. In the case of small-volume BM, in our study, less than 0.04 cc, the detection sensitivity was 85.7%. In detection performance analysis, we could get TP, FP, and FN; however, patients in all our test-set contain at least one BM, we could not define true negatives (TN) as a prediction for the patients with no BM. In addition, since our final results detects BM by merging cropped patches, it is practically impossible to obtain TP, FP, FN, and TN for each cropped patch. Therefore, TN cannot be defined as a correctly identified healthy patch. In the end, we used sensitivity and average false positive rate to measure detection performance.

Blood vessels, as well as metastatic lesions, demonstrate high signal intensity on T1Gd; therefore, normal blood vessels are occasionally mistaken for metastatic lesions, and the detection of metastatic lesions can be affected by small vessels. Detection of BM with black-blood (BB) imaging, complementary to contrast-enhanced 3D gradient-echo, has therefore garnered attention [31]. Jun, Y. et al. [32] reported the results of BM detection using deep-learned 3D BB imaging with 3D CNNs. The diagnostic performance of radiologists was 0.9708 with deep-learned BB and 0.9437 with original BB imaging, showing that BB imaging combined with DL algorithms is highly comparable to the original BB imaging. Therefore, our model or other DL-based BM segmentation and detection models could be trained and tested using BB imaging or other MR image sequencing.

The strength of our study is that we included relatively small-volume BM. As shown in Table 4, the median volume of our dataset is smaller than that of the previous studies. To our best knowledge, our study included BM with the smallest volumes. Although the proposed model showed a rather low DICE, this result is mostly due to omitting the slices at the beginning and end of BM. However, there is no need to be discouraged by this results because these results can be easily corrected by radiation oncologist and it provides a margin for delineation when performing SABR. In contrast to the segmentation performance, the detection performance of the proposed model is outstanding despite the relatively small-volume BM. Unlike our study using only one MRI sequence (T1Gd), some previous studies have also used T2-weighted or FLAIR to detect BM. However, there is a limit to simultaneously obtaining multiple MRI modalities in clinical practice. The DL model of the previous studies has the downsampling layer as ours, after downsampling, low-resolution feature information is transmitted to the next convolution layer. In this process, a small-volume BM could be omitted. To compensate for this, we applied the 2.5D overlapping patch technique, which can better localize small volume BM. As a result, the proposed model was able to detect small BM well. Thus, our results are more clinically useful than those of the previous studies and may help radiation oncologists in choosing the treatment modality for BM, whole-brain RT, or SABR.

In clinical practice, detection of small-volume BM is necessary for medical oncologists, radiologists, and neurosurgeons to choose the appropriate treatment modality. Generally, whole-brain RT is considered the treatment of choice for numerous BM. However, from a radiation oncologist’s perspective, with advances in RT technique and more frequent use of SABR, precise segmentation of all BM is essential to choose an appropriate treatment modality. Considering that it is difficult to obtain assistance from radiologists for the segmentation of all BM, our DL model would help radiation oncologists with the autosegmentation of BM.

There are several limitations to this study. Our study reduced false positives through the overlapping patch technique and 2.5D training, but false positives still existed. If the number of false positives is high, a labor-intensive process is required to screen them out. There are many methods to reduce false positives, but it is important to check where false positives occur. We summarized it in a few cases and propose solutions that should be applied to further studies. Solutions to reduce the occurrence of false positives include sophisticated skull-stripping application [33], extensive gamma correction application, and additional black-blood sequencing.

## 5. Conclusions

In this study, we have proposed the DL model, in which applied 2.5D overlapping patch technique to detect and segment BM even for small-volume BM. The proposed model could detect all but some omissions of <0.04 cc BM and segmentation with good performance. Our results suggest considerable benefits for gross tumor volume detection and segmentation for stereotactic radiosurgery or SABR by assisting the radiation oncologist with good performance. 

## Figures and Tables

**Figure 1 cancers-14-02555-f001:**
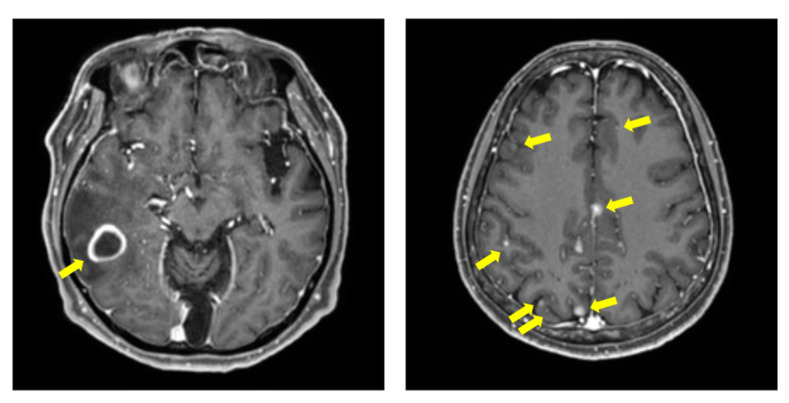
Examples of T1Gd images with BM. The (**left**) image shows the randomly selected samples of large-volume BM. The (**right**) image shows the randomly selected samples of small-volume BM with multiple metastases. In each images, yellow arrows indicate the BM.

**Figure 2 cancers-14-02555-f002:**
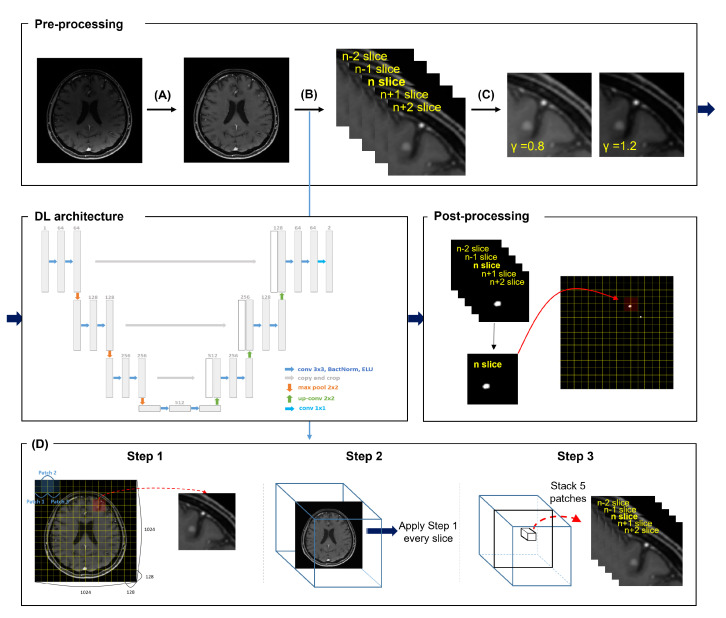
2D U-Net with preprocessing and postprocessing, which is effective for small volumes. Preprocessing composed of bias field correction (**A**), 2.5D overlapping patch technique (**B**), and random gamma correction (**C**) is applied to the MR image. Postprocessing is applied to prediction through 2D U-Net to obtain results. By cropping the 1024 × 1024 size image by overlapping the 128 × 128 size patch, it was effective for small volumes, and by adding 2 slices in each above and below the reference slice, 5 channels are configured to reflect volume information (**D**). In step 1, form a cropped patch by sliding a 64 × 64 size patch from left to right and top to bottom of the slice. In step 2, apply the process of step 1 to all slices of the patient individually. Finally, in step 3, stack the patches with the same x-y coordinates in 5 slices along the z-axis.

**Figure 3 cancers-14-02555-f003:**
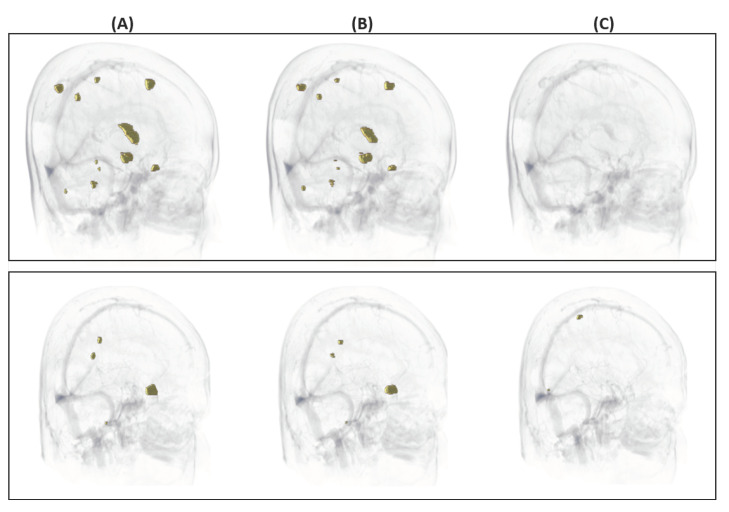
Three-dimensional visualization of the results. The first row represents an example of a case where false positives were not found, and all 11 BM were detected. In the second row, all 4 BM were detected, but 2 false positives were found. Each column represents the ground-truth gross tumor volume delineated manually (**A**), true positives from prediction (**B**), and false positives from prediction (**C**).

**Figure 4 cancers-14-02555-f004:**
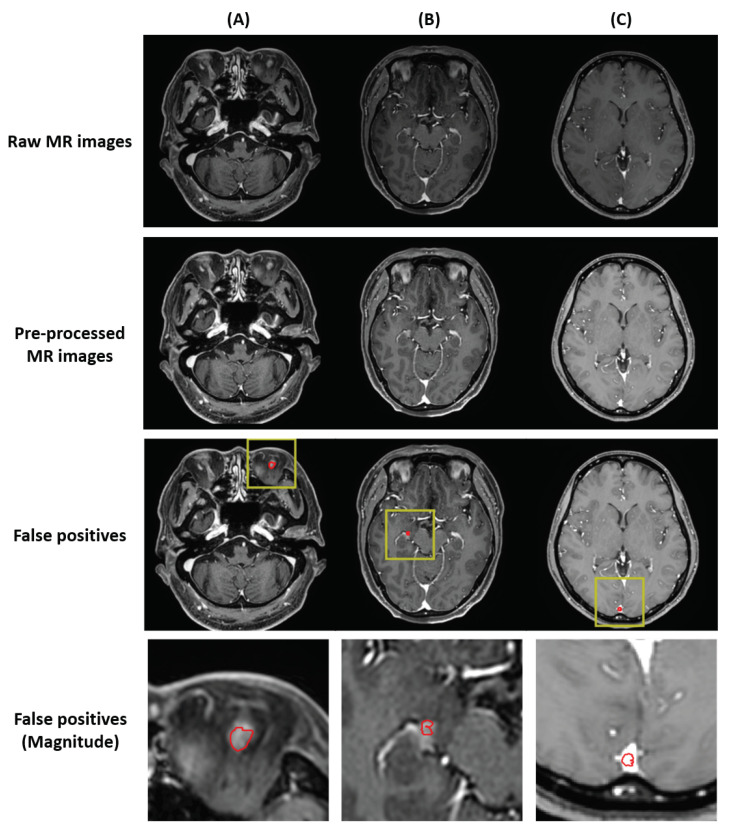
Locations of the false positives. (**A**) Delineation created outside the brain; can be solved with skull-stripping. (**B**) Delineation created in a structure with high intensity; can be solved with extensive gamma correction. (**C**) Delineation created in the superior sagittal sinus; can be solved with black-blood sequencing. In each images, red in the yellow bounding box indicates false positives.

**Figure 5 cancers-14-02555-f005:**
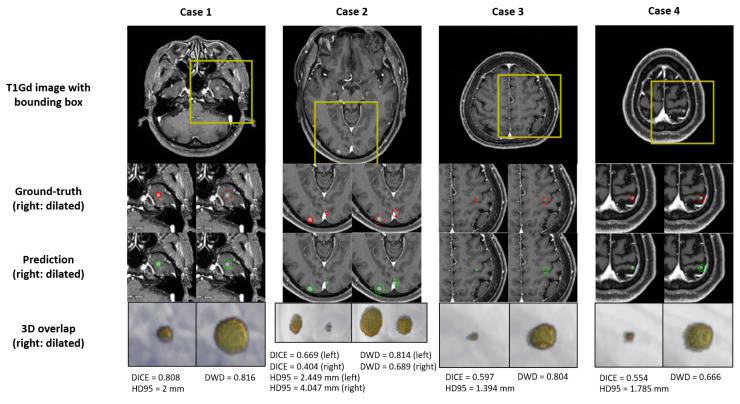
Examples of the ground-truth delineation and predicted delineation. The yellow bounding box in the first row indicates the area where false positives occurred. The second row shows the manual delineation of the tumor volume and the manual delineation dilated to 3 mm. The third row shows the predicted delineation from DL and the predicted delineation dilated to 3 mm. For the second and the third rows, red and green indicate manual and predicted delineation, respectively. The last row shows the 3D rendering overlapped with manual delineation and predicted delineation, and 3D rendering dilated to 3 mm each. In the last row, renderingRendering in red is predicted delineation and rendering in yellow is manual delineation.

**Table 1 cancers-14-02555-t001:** Patient Characteristics.

Variables	Total (65)	Train (+Valid) Set (53)	Test Set (12)	*p* Value
Age (years)				
Median (range)	63 (19–87)	63 (19–87)	63 (26–81)	0.869
Sex				1
Male	35 (54)	29 (55)	6 (50)	
Female	30 (46)	24 (45)	6 (50)	
Primary cancer				0.604
Lung	56 (86)	45 (85)	11 (92)	
Breast	4 (6)	4 (7)	-	
Others	5 (8)	4 (8)	1 (8)	
Total number of BM	603	545	58	
>0.04 cc	458 (76)	414 (76)	44 (76)	
≤0.04 cc	145 (24)	131 (24)	14 (24)	
Volumes of BM				
Max	67.426	67.426	1.219	
Min	0.02	0.02	0.021	
Median	0.074	0.074	0.068	
Mean	0.552	0.592	0.158	

**Table 2 cancers-14-02555-t002:** Deep Learning Performance in Test-Set Patients according to Metastases.

No. of BM Per Patient	No. of Patients	Sensitivity [%]
≥10	3	93.5
<10	9	100
**Total**	12	96.6

**Table 3 cancers-14-02555-t003:** Summary of Detection and Segmentation Performance.

Volume (cc)	No. of BM	Sensitivity [%]	No. of FPs	DICE	DWD	HD95 [mm]
>0.1	24	100	4	0.64	0.8	2.502
≤0.1	34	94.1	11	0.48	0.72	1.724 *
0.08–0.1	1	100	2	0.82	0.9	1
0.06–0.08	10	100	2	0.53	0.78	1.979
0.04–0.06	9	100	2	0.56	0.75	1.608
0.02–0.04	14	85.7	5	0.38	0.63	1.689 *
**Total**	58	96.6	15	0.55	0.75	2.057

* HD95 was calculated except for BM that were not detected.

**Table 4 cancers-14-02555-t004:** Comparison between our study and other studies.

Authors	Median Vol. of BM [cc]	Sensitivity [%]	Avg. No. of FPs	DICE
Losch, M. et al.	NA	82.8	7.7	0.66
Charron, O. et al.	0.5	93	4.4	0.79
Xue, J. et al.	2.22	96	NA	0.85
Grovik, E. et al	NA	83	3.4	0.79
Dikici, E. et al.	0.16	90	9.12	NA
Bousabarah, K. et al.	0.31 (train)/0.47 (test)	NA	NA	0.71
Our study	0.074 (train)/0.068 (test)	96.6	1.25	0.55 ${}^{a}$

^*a*^ DWD is 0.75 and HD95 is 2.057 mm.

## Data Availability

The original contributions presented in the study are included in the article. Further inquiries can be directed to the corresponding authors.

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
