# Peer review of "Deep-Learning-Based Automatic Detection and Segmentation of Brain Metastases with Small Volume for Stereotactic Ablative Radiotherapy"

_cancers, 2022, doi:10.3390/cancers14102555_

Round 1

Reviewer 1 Report

The manuscript submitted by  Yoo and co-authors describes a segmentation and evaluation of brain metastasis that have been observed with MRI. The manuscript presents the methodology, obtains accuracy metrics and illustrates the process. The manuscript is interesting, fairly well written and should be of interest to the community. However, there are several factors that should be properly addressed in order to be recommended for publication.

  • The manuscript should indicate from the abstract how the results obtained compared with previous literature, if the manuscript outperforms other methodologies then it deserves to be published, and if it does not outperforms, there may be reasons that will be addressed in the document but simply claiming a considerable benefit is not enough.
  • The manuscript should be proofread carefully, some acronyms are used before they are defined, English could be improved.
  • The introduction needs to be revised. For a start claiming something is becoming popular with two references from 2000 and 2013 is hardly a good argument. As the introduction may be read by people who are neither radiologists nor computer scientists it should provide more details. Specifically, many readers will not be familiar with what is considered a large or a small volume of a brain metastasis. This should be clearly presented and illustrated with figures. In the same tone, previous studies of brain metastasis with/without deep learning should be discussed so that the contribution of this work is clear by the end of the introduction.
  • Materials and methods needs to introduce the materials to the reader with some images! We have to wait until the results to see how a BM looks like. Some details are confusing: how was the pixel spacing adjusted to 0.195? I think that what the authors mean by this is that they resampled the image to fit this resolution. The whole concept of 2.5D is not clearly explained (and is used before it is defined). 2D is clear 3D is clear but 2.5 can be interpreted in many ways, and “stacked by stacking” is a rather bad definition. Top and bottom slices normally refer to the Top and the Bottom slices of a whole stack. I suspect the authors do not mean these slices were used.
  • Figure 1 is interesting but it is just TOO SMALL! I suggest that the authors print this page in A4 or letter and try to read it. I struggled with a 27 in Monitor.
  • In the same way that the authors defined sensitivity, they should define all other metrics.
  • The justification to use DWD is not really necessary. It is well known that Dice and Jaccard are very strict with small objects. The authors may want to calculate accuracy as an extra metric.
  • Results, I cannot think of a reason why the authors sent the table of results to supplementary materials other than the results were not what they wanted to show. A single table like this has to be in the main body of the document and then it should be discussed why their method had the lowest Dice but a decent sensitivity and average number of FP (why not use specificity or accuracy?)
  • From Figures 3 and 4 it is not possible to assess how good or bad are the results, these are tiny! Even at 400% in a 27 in monitor. The authors should include zooms to the regions of interest and show the TP/TN/FP/FN of some metastasis to be able to appreciate if the method is good or not.
  • The discussion should be improved, for example “Because we applied effective pre-processing including 2.5D overlapping patch technique, it is effective even in small-volume BM;” this is a circular argument, and what does effective mean?

Reviewer 2 Report

The paper titled “Deep learning-based automatic detection and segmentation of gross tumor for stereotactic ablative radiotherapy in small-volume brain metastases” deals with an automatic detection and segmentation approach to brain metastases (BM) using advanced deep learning (DL) algorithms. The authors used an N-Net model for the detection and segmentation of small-volume BM. The paper has some shortcomings, and they must be corrected before the acceptance of the paper. My observations are the following:

  • Abbreviate BM in summary, where it appears first
  • Remove the repetition in the summary and abstract parts.
  • Define BMDS?
  • The author must add the recently reported works in the introduction part. The introduction section is too short.
  • The introduction section must conclude with a research gap, and the authors also add some detail about their work.
  • Define the “n” before reporting the values in Section 2.1.
  • It is better if the authors define all the parameters used to describe the dataset for the readers who don’t have the related background (sections 2.1 and 2.2).
  • The authors must add the link for the dataset repository. So, the readers can also use that dataset to replicate the results.
  • Section 2.3 is very briefly written; the authors must explain the work and the reason for each step in detail.
  • The authors apply the under-sampling technique to balance the data, and the authors must mention the final sample values of the dataset.
  • Why the images are cropped to 128 × 128 sizes. Is this the optimal value? If yes, then why?
  • Why did 64 pixels use to form a cropping patch?
  • If the overlapping patch technique does not assume that the entire tumor will be contained within each patch. Then what is the advantage to make these patches?
  • After applying the under-sampling technique, the authors again apply the data augmentation approach before training. why?
  • In my opinion, it is better to upgrade the data in the first step.
  • The authors used the U-Net model in their work, the model is already presented in the literature? What is the novelty of this work?
  • The information presented related to the use of the machine must not be present in the Deep learning details section.
  • In the results section, the authors must add complete details of the size of the training vector, how many images are used for training the model, and how many are used for testing the model.
  • The authors must add the confusion matrix for a better presentation of the results.
  • The authors must improve the discussion section, there is a lot of repetition in it. This section must focus on the outcomes and novelty of the work compared to the literature.
  • It is better if the authors add the comparison table of their proposed literature.
  • It is better if the authors add a conclusion section at the end.
  • The paper must be proofread by a native English speaker.

Round 2

Reviewer 2 Report

The authors address all my comments. The paper can be accepted for publication.